# CRISPR/Cas9-Mediated *SlMYBS2* Mutagenesis Reduces Tomato Resistance to *Phytophthora infestans*

**DOI:** 10.3390/ijms222111423

**Published:** 2021-10-22

**Authors:** Chunxin Liu, Yiyao Zhang, Yinxiao Tan, Tingting Zhao, Xiangyang Xu, Huanhuan Yang, Jingfu Li

**Affiliations:** College of Horticulture and Landscape Architecture, Northeast Agricultural University, Harbin 150030, China; liuxx130@sina.com (C.L.); zyy553206@sina.com (Y.Z.); 186153xx@sina.com (Y.T.); zhaottneau@gmail.com (T.Z.); xuxyneau@gmail.com (X.X.)

**Keywords:** *Solanum lycopersicum*, *SlMYBS2*, gene expression, disease resistance, *P. infestans*

## Abstract

*Phytophthora infestans* (*P. infestans*) recently caused epidemics of tomato late blight. Our study aimed to identify the function of the *SlMYBS2* gene in response to tomato late blight. To further investigate the function of *SlMYBS2* in tomato resistance to *P. infestans*, we studied the effects of *SlMYBS2* gene knock out. The *SlMYBS2* gene was knocked out by CRISPR-Cas9, and the resulting plants (*SlMYBS2* gene knockout, *slmybs2-c*) showed reduced resistance to *P. infestans*, accompanied by increases in the number of necrotic cells, lesion sizes, and disease index. Furthermore, after *P. infestans* infection, the expression levels of pathogenesis-related (*PR*) genes in *slmybs2-c* plants were significantly lower than those in wild-type (AC) plants, while the number of necrotic cells and the accumulation of reactive oxygen species (ROS) were higher than those in wild-type plants. Taken together, these results indicate that *SlMYBS2* acts as a positive regulator of tomato resistance to *P. infestans* infection by regulating the ROS level and the expression level of *PR* genes.

## 1. Introduction

During their growth, plants are often subjected to biotic and abiotic stresses [1,2], such as disease, high and low temperatures, drought, salinity, and other adverse events. To withstand various adverse environments, plants undergo specific gene transcription and expression changes. Transcription factors are regulatory proteins that bind to corresponding cis-acting elements and regulate gene expression [3]. MYB transcription factors are a large and versatile plant transcription factor and are widely involved in the physiological and biochemical reactions of plants, and they play key roles in plant growth and development [4]. MYB transcription factors are also closely related to the biotic and abiotic stress responses of plants. The first MYB transcription factor was identified in avian leukosis virus and named v-myb avian myeloblastosis viral oncogene homolog [5]. Paz-Ares et al. (1987) were the first to clone an MYB transcription factor that regulated the synthesis of maize anthocyanins [6].

Further research revealed an increasing number of MYB transcription factors involved in biotic and abiotic stress. *AtMYB34*, *AtMYB44*, *AtMYB51*, *AtMYB75*, and *AtMYB102* in Arabidopsis effectively defend against herbivorous insects [7,8]. *MYB30* and *MYB96* respond to pathogen stress by regulating the expression of corresponding PR genes in Arabidopsis [9,10]. *AtMYB12*, *AtMYB60*, *AtMYB75*, and *AtMYB96* can adjust the drought resistance mechanism of *Arabidopsis thaliana* to adapt to drought conditions [11,12,13]. The overexpression of *OsMYB55* in rice increased the total amino acid content in the plant, which improved plant tolerance to high temperature. *OsMYB55* plays a certain role under high-temperature stress [14]. *OsMPS* in rice is a 2R-MYB transcription factor that improves the salt tolerance of plants by mediating the synthesis of plant hormones and cell walls [15]. As a negative regulator, *AtMYB44* in Arabidopsis regulates salt tolerance [16]. MYB transcription factors are also widely involved in low-temperature stress responses in plants, such as *AtMYB15* in Arabidopsis [17], *GmMYB76* and *GmMYB1* in soybeans [18], and *MdMYB88* and *MdMYB124* in apples [19].

Tomato (*Solanum lycopersicum*), a fruit vegetable, is widely cultivated worldwide. However, the epidemic of late blight has adversely affected the global production of tomatoes [20]. Late blight is an oomycete disease, and late blight caused by *Phytophthora infestans* (*P. infestans*) infection of leaves, stems, and fruits causes disease spots and may kill the entire plant; however, the pathogen also infects tomato seeds [21,22,23]. The total production loss of fresh and processed tomatoes due to tomato late blight reached USD 40 to 60 million in the US in 2009 [24]. An outbreak of late blight may cause large-scale damage within a few days. Low-level late blight is also difficult to detect, and controlling blight spread with sprayed pesticides is difficult after obvious symptoms appear. New strains that are resistant to antimicrobial agents have also been identified [25,26]. Therefore, it is necessary to formulate effective strategies for preventing and controlling late blight. Because tomatoes are such a high-value crop, late blight is a current research hotspot. The development of biotechnologies has deepened the research scope, which includes tomato growth and physiological and biochemical reactions at the molecular biology level. However, relatively few studies have investigated tomato MYB transcription factors. Fourteen MYB-related gene fragments were first cloned in tomato in 1996, but their functions have not been fully elucidated [27]. The *b1* gene of the R2R3-MYB family, related to the meristem, was identified in tomato in 2002 [28], and MYB family genes that regulate the synthesis of tomato anthocyanins were discovered in 2003 [29]. An increasing number of MYB genes related to biotic stress have been found in tomato in recent years. One study revealed that the overexpression of *OsMYB4* in tomato increased resistance to mosaic virus (ToMV) [30]. Twenty-four R2R3-MYB transcription factors in the tomato genome were identified based on their association with Arabidopsis R2R3-MYBs. After *Phytophthora* infection, the expression of *MYB49* was increased significantly, and high resistance to *P. infestans* was observed in tomato plants with *OsMYB4* overexpression. Moreover, the number of necrotic cells and the lesion sizes were decreased [31]. To improve resistance of tomato plants to yellow leaf curl virus (TYLCV), virus-induced gene silencing of *SlMYB28* has been used [32].

The genome engineering process is a critical and dynamic technique that has been used in recent years to study plant function. CRISPR-Cas9 won a groundbreaking award in plant biology “Methods of the Year” in 2011. An effective CRISPR-Cas9 type II system was recently reported in various plants, such as rice [33,34], wheat [35,36], cabbage [37], lactuca [38], and cucumber [39], and CRISPR-Cas9 technologies are increasingly being used for tomato. Li et al. (2018) used CRISPR/Cas9 technology to edit five genes that improved the habitat of tomato plants, advanced the flowering time, and increased the size of tomato fruit and the content of vitamin C in the tomatoes [40]. The flowering suppressor gene SELF-PRUNING 5G (SP5G) in tomato has been knocked out by CRISPR/Cas9, which induced earlier flowering and yield of the edited seedlings [41].

Because the information on genes resistant to tomato late blight is limited, the identification of higher resistance genes to late blight is urgent. The role of *SlMYBS2* in tomato late blight has not been reported, and we previously showed that *SlMYBS2* was induced by a late blight pathogen. Therefore, the present study aimed to determine the functional impact of the *SlMYBS2* gene during *P.*
*infestans*-induced tomato late blight and is the first to clarify the role of *SlMYBS2* in this disease. Exploring the induced expression changes and elucidating the function of *SlMYBS2* in tomato disease resistance are very important, and this research provides candidate genes for tomato disease-resistance breeding, making it possible to obtain disease-resistant tomato varieties.

## 2. Results

### 2.1. Phylogenetic Analysis of SlMYBS2

To identify putative R2R3MYB proteins in tomato, we performed a BLASTP search against the tomato genome database (http://mips.helmholtzmuenchen.de/plant/tomato/searchjsp/blast.jsp (accessed on 10 January 2020)) using 126 R2R3MYB protein sequences in Arabidopsis and the hidden Markov model (HMM) profile of the MYB-binding domain as queries. Finally, 121 typical R2R3MYB genes were confirmed by the Pfam and SMART programs. An unrooted NJ phylogenetic tree was generated based on the alignment of the corresponding tomato R2R3MYB protein sequences. For statistical reliability, we conducted bootstrap analysis with 1000 replicates. The 121 members of the SlR2R3MYB family were subdivided into 16 subgroups, designated A–P, according to clades with at least 50% bootstrap support (Figure 1A).

Based on our previous pathogen-induced transcriptome analysis, we selected five genes of the MYB family that were significantly induced. RT-qPCR analysis showed that late blight significantly induced the *SlMYBS2* gene (Appendix A), the function of which has not yet been reported; thus, we focused on *SlMYBS2* in this study. *SlMYBS2* contains 267 amino acids, with an MYB domain in the middle (Figure 1A). This gene was expressed at the highest levels in leaves, as determined by the public tomato eFP browser tool (http://bar.utoronto.ca/eplant_tomato/ (accessed on 20 February 2020)) (Appendix A).

To reveal the evolutionary relationships between *SlMYBS2* and other MYB homologs, 17 predicted MYB proteins from four species were included in a phylogenetic analysis. Tomato, Arabidopsis, and rice were placed in a single group, suggesting that they share a common origin. In addition, *SlMYBS2* was located in the same cluster as *SlMYB49* in tomato, thus suggesting that *SlMYBS2* and *SlMYB49* have similar functions.

### 2.2. Subcellular Localization Analysis

Subcellular localization prediction results in Cell-PLoc2.0 showed that the protein most likely functions in the cell nucleus, followed by the cell membrane. To determine the subcellular localization of *SlMYBS2*, a chimeric gene expression cassette containing a *SlMYBS2*-GFP fusion gene under the control of the 35S promoter was expressed in the leaves of *A. thaliana* and *Nicotiana benthamiana* (Figure 2A). We found that *SlMYBS2*-GFP signals were present in the nucleus only (Figure 2B,C), which was in agreement with its role as a transcription factor. As a control, we also examined the subcellular localization of the GFP in leaf cells, and green signals were obviously present in both the cytosol and nuclei (Figure 2B,C).

### 2.3. SlMYBS2 Is a Transcription Factor with Transcriptional Activity 

We analyzed the transcriptional activity of *SlMYBS2* using a yeast system to investigate its potential role. On tryptophan-deficient medium (SD/−Trp), both the experimental and control groups grew normally. On tryptophan, adenine, and histidine (SD/−Trp−Ade−His) medium and the same medium supplemented with 5-bromo-4-chloro-3-indole-ad-galactoside (X-a-gal; SD/−Trp−Ade−His+X-a-gal), the control group did not grow normally, while the experimental group grew normally and colonies turned blue (Figure 2D). These results indicate that *SlMYBS2* has a transactivational capacity.

### 2.4. Expression of SlMYBS2 in Response to Defense Signaling-Related Hormones

We also examined the dynamics of *SlMYBS2* expression in tomato plants after treatment with SA and MeJA, two defense signaling-related hormones, and the results are shown in Figure 3. The spray application of exogenous SA to AC tomato plants promoted the expression of *SlMYBS2*, and the expression of *SlMYBS2* peaked at 16-fold higher than the initial level at 3 h (Figure 3). After the spray application of exogenous MeJA, the expression of *SlMYBS2* peaked at 3-fold higher than the initial level at 6 h (Figure 3). These data indicate that tomato *SlMYBS2* responds to SA and JA, two well-known defense signaling-related hormones, with differential patterns of expression.

### 2.5. Knocking out SlMYBS2 Resulted in Reduced Resistance to P. infestans

To further identify the function of *SlMYBS2*, we mutated *SlMYBS2* in the Alisa Craig strain via the CRISPR/Cas9 system. The two target sequences selected were located in the first exon (Appendix A), and a schematic diagram of the resultant vector is depicted in Appendix A. Using the method shown in the figure, the desired transgenic tomato plants were obtained. As a result, four kanamycin-resistant tomato lines were obtained from the T0 transgenic lines. Sequencing analysis and DSDecode (http://www.ygliulab.club/dsdecode/ (accessed on 11 June 2020)) decryption revealed that two of the lines *(slmybs2-c-1* and *slmybs2-c-2*)) had mutations at the targeted genes, although all of the mutations were heterozygous. We identified a total of 36 T1 plants (16 *slmybs2-c-1* and 20 *slmybs2-c-2*) and obtained four types of *SlMYBS2* mutations (1 bp (A) insertion and 1 bp (C) deletion) from the self-pollinated T0 lines (Appendix A). Even though two sgRNAs were selected to target the *SlMYBS2* gene, all of these mutations occurred at the second target site. In addition, the two target sequences of *SlMYBS2* were BLAST searched against the tomato genome with a low E-value (1 × 10^−^^1^) to identify potential off-target sites. No sites other than the targets were found in the tomato genome, indicating that no off-target events would occur in mybs2 mutants. Plants with homozygous *SlMYBS2* mutations (*slmybs2-c-1* and *slmybs2-c-2*) but without the T-DNA insertion were chosen to generate T2 plants for further analysis.

To first determine the role of *SlMYBS2* in the defense against *P. infestans*, the *slmybs2-c*- and AC plants were infected with *P. infestans*, and their physical appearances were assessed 5 days later. However, the lesions on leaves from *slmybs2-c*-infiltrated plants were significantly larger at 5 dpi (Figure 4A), as they were approximately 34.4% larger than those in AC plants (Figure 4B). The AC plants exhibited a strong HR at 5 dpi with *P. infestans*, as determined by trypan blue staining. In contrast, no visible HR was observed in the *slmybs2-c-* plants at 5 dpi; the hyphae gradually grew, and the lesions were aggravated and transparent. In contrast with those of the AC plants, the leaves of the *slmybs2-c*- plants were sensitive to *P. infestans* infection. These data demonstrate that knock out of *SlMYBS2* resulted in reduced resistance to *P. infestans*; thus, both *SlMYBS2* proteins are required for resistance to *P. infestans.*


### 2.6. ROS Accumulation Was Increased in Slmybs2 Mutant Plants Compared with Control Plants after Inoculation with P. infestans

To elucidate the possible mechanism underlying the reduced resistance of *slmybs2-c*- plants, we analyzed and compared the accumulation of ROS after infection with *P. infestans* between *slmybs2-c-* and AC plants. No difference in the accumulation of H_2_O_2_ and O^2−^, as detected by DAB and NBT staining, was observed in the leaves of AC and *slmybs2-c* plants without *P. infestans* infection (Figure 5), indicating that knocking out *SlMYBS2* itself did not affect the generation or accumulation of H_2_O_2_ and O^2−^ in tomato plants. After infection with *P. infestans*, significant accumulation of H_2_O_2_ and O^2−^, shown as brown and blue precipitates, respectively, was detected in the leaves of the *slmybs2-c*-*1*-, *slmybs2-c*-*2*-, and AC-infiltrated plants. However, the intensity of the stained areas was consistently increased in the leaves of the *slmybs2-c*-*1*- and *slmybs2-c*-*2*-infiltrated plants compared with the AC-infiltrated plants after infection with *P. infestans* (Figure 5). These data indicate that knocking out *SlMYBS2* accelerated the generation and accumulation of H_2_O_2_ and O^2^^−^ upon infection with *P. infestans*.

### 2.7. Knocking out SlMYBS2 Affected the Expression of Scavenging-Related and Defense-Mediated Genes after Infection with P. infestans

To explore the possible mechanism underlying the increased accumulation of H_2_O_2_ in *slmybs2-c-* plants, we analyzed and compared the expression levels of genes encoding catalases (CATs), superoxide dismutases (SODs), and ascorbate peroxidases (APXs) in *slmybs2-c*- and AC-infiltrated plants. As shown in Figure 6A, the expression levels of these genes in the leaves of *slmybs2-c*-infiltrated plants were generally decreased. These results suggest that ROS accumulation in *slmybs2-c*-infiltrated plants was potentially due to the decreased expression of ROS scavenging genes.

We next analyzed the expression changes in defense-related genes regulated by the JA- and SA-mediated signaling pathways to explore the possible molecular mechanism underlying the reduced disease resistance in *slmybs2-c*- plants. After *P. infestans* infection, the expression levels of PR genes in *slmybs2-c*- plants were considerably lower than those in AC plants. These data demonstrate that knocking out *SlMYBS2* attenuated the defense response in tomato upon infection with *P. infestans* by affecting the expression of defense-related genes that are regulated by the JA/SA-mediated signaling pathway.

## 3. Discussion

### 3.1. CRISPR/Cas Technology Can Be Used to Develop Disease-Resistant Tomatoes

Different from traditional gene editing techniques, the CRISPR/Cas9 system can edit multiple gene loci simultaneously, which greatly improves the efficiency of gene editing. It has been recognized as an important means for enhancing crop disease resistance [42,43,44]. Analysis of the mutant plants constructed by the CRISPR/Cas9 system showed that the gene was resistant to a variety of plant pathogens, and the editing of the gene had no significant effect on plant growth, which proved that the technique could be used to cultivate disease-resistant tomato [45]. In wild tomato, *Mlo* (mildew, resistant locus) encodes a membrane protein that is very sensitive to powdery mildew. CRISPR/Cas9 was used to delete this gene, and the mutant plants were found to have complete resistance to powdery mildew [46]. The pathogenic tomato variant *Pseudomonas syringae* can produce corontin, which can stimulate the stomatal opening of tomatoes and facilitate bacterial infection that produces bacterial spot disease. Using CRISPR/Cas9 to knock out *SlJAZ2* gene, it was found that the stomatal opening caused by corononins could be prevented, thus rendering the tomato resistant [47]. In this study, *SlMYBS2* was edited by CRISPR/Cas, and two mutant plants were obtained. Compared with wild-type plants, mutant plants were less resistant to late blight. This provided a new idea for breeding tomato resistant to late blight.

### 3.2. The SlMYBS2 Transcription Factor Resists P. infestans through the SA and JA Signaling Pathways

A defense system composed of various signal transduction pathways exists in plants to resist biotic and abiotic stresses, and plant hormones and their signal transduction pathways play key roles in disease resistance responses [48,49]. SA and JA are important signaling molecules in plant innate immune responses (PTIs, ETIs) and regulate plant disease and abiotic stress responses. *NPR1* is located downstream of SA biosynthesis; it has been shown that *NPR1* is an SA-responsive transcriptional coactivator [50]. SA-mediated *OsNPR1* gene expression plays a key role in rice disease resistance [51]. Exogenous application of benzothiadiazole (BTH) upregulates the expression of the *OsWRKY45* gene and improves the resistance of rice to *Magnaporthe oryzae* [52,53]. JA enhances the resistance of plants to plant pathogenic fungi, bacteria, and viruses in many ways. For example, JA was shown to enhance the resistance of wheat to powdery mildew [54,55]. JA increased the expression of the *SlPR3* gene to enhance the resistance of tomato plants to pathogens [56]. Other studies showed that exogenous SA also induced MYB transcription factor responses. For example, the expression of the *NtMYB1* transcription factor was increased in tobacco when active SA was administered externally, which thereby increased the expression of disease-related proteins that significantly improved tobacco disease resistance [57]. In this experiment, the expression of *SlMYBS2* was analyzed by qRT-PCR after the exogenous application of SA and JA, revealing increased expression in tomato leaves. In addition, in this study, the expression levels of PR genes in gene knockout plants were markedly inhibited. In general, tomato *PR1*, *PR2*, and *PR5* are SA-dependent genes, and their expression leads to enhanced resistance to hemibiotrophic trophic pathogens [58,59], whereas *PR3* is a marker gene for JA signaling whose expression enhances the resistance of plants to dead-nutrient pathogens [60]. Therefore, we speculate that the *SlMYBS2* transcription factor resists *P. infestans* through the SA and JA signaling pathways.

### 3.3. Knocking down SlMYBS2 Weakened Resistance to P. infestans

Expression of *MYB49* in tomatoes revealed that tomato plants overexpressing *MYB49* had high resistance to *P.*
*infestans*. The number of necrotic cells and the size of disease spots were decreased [31]. Compared with our experiment, we used the CRISPR/Cas system to knock out the *SlMYBS2* gene and observed the phenotype and found that the resistance of the mutant plants was weakened. We found that the mutant plants had weakened resistance, increased necrotic cells, and enlarged lesions. These results were similar to those of the above test, and both genes play a positive role in disease resistance. These results are similar to those herein and indicate that the two genes have similar functions.

Pathogen infection may activate different plant defense pathways [61] and is often accompanied by the production of ROS, which play key roles in the plant defense response [62]. The ROS burst in response to biotic stresses has a protective role, as evidenced by studies showing that some ROS function as secondary messengers of signal transduction pathways controlling pathogen defense responses [63]. However, because excessive ROS cause serious damage, plants and animals tightly regulate ROS production and detoxification [64]. ROS produced in the late stage of infection can have toxic effects on plant cells, resulting in lipid peroxidation, cell membrane damage, pathogen susceptibility, and cell death [65,66]. The ROS scavenging mechanism can protect plants from diseases by increasing the expression level of ROS scavenging-related genes and the activity of antioxidant enzymes [64]. It has been shown that MYB transcription factor in *Arabidopsis thaliana* and tomato can enable plants to gain tolerance to both biological and abiotic stresses by activating antioxidant defense mechanisms [31,64]. We also explored whether *SlMYBS2* was involved in this protection mechanism. By DAB and NBT staining, it was found that the content of ROS accumulation in leaves of *slmybs2-c* plants after pathogen infection was significantly higher than that of control plants. qRT-PCR quantitative detection showed that *SlMYBS2* knockout reduced the expression of ROS scavenging-related genes. These results suggest that the reduced disease resistance of *SlMYBS2* plants may be related to the decreased expression level of ROS scavenging-related genes in plants.

### 3.4. Model for the Putative Role of SlMYBS2 in the Regulation of Pathogen Defense Responses

The above results indicate that *slmybs2-c-* plants accumulated more hydrogen peroxide and oxygen anions than AC plants after *P. infestans* infection (Figure 7). *PR* expression was lower in *slmybs2-c-* plants than in AC plants. *SlMYBS**2* knockout weakened the resistance of tomato to *P. infestans*, which may have been caused by the increased accumulation of ROS and the inhibited expression of *PRs*.

## 4. Materials and Methods

### 4.1. Plant Growth, Hormone Treatment, and Disease Assays

Tomato (*Solanum lycopersicum* cv. Ailsa Craig) including wild type and two mutant type (slmybs2-c-1, slmybs2-c-2), *Nicotiana benthamiana*, and *Arabidopsis thaliana* were grown on sterilized nutrient soil in a light incubator with 50–60% relative humidity (RH) under 16 h of light at 24 °C and 8 h of dark at 16 °C. The light intensity was 500 μmol m^−2^ S^−1^ of photosynthetic photon flux density (PPFD).

Phytophthora was isolated from the surfaces of infected tomato leaves. *P. infestans* was incubated on rye medium and cultured for 10 days at 21 °C. A conidiophore was selected and incubated in liquid rye medium for 8 days. Spore suspensions of the strain were filtered through 4 layers of gauze. A spore suspension of *P. infestans* (1 × 10^6^ spores/mL of water) was used to inoculate detached tomato leaves (six leaves of control plants and mutant plants) and tomato plants (fifteen control and mutant plants) were infected by spraying. The infected plants were cultured under a light incubator with a light-dark (LD) cycle of 16 h L:8 h D, a light intensity of 500 μmol m^−^^2^ S^−^^1^ of PPFD, a temperature of 21 °C, and an RH of 60%. The disease status of the plants was observed daily after inoculation, and leaves were harvested at 0–5 days post-inoculation (dpi) for further analysis.

The tomato plants were sprayed with the same amount (5 mL) of 400 μM salicylic acid (SA) and 200 μM methyljasmonate (MeJA) individually at 4 weeks of age. For analysis of gene expression, the leaves were collected at 0, 3, 6, and 12 h after hormone treatment. Each treatment group contained 10 plants, and the entire experiment was repeated 3 times.

### 4.2. Gene Cloning and Bioinformatics Analysis

The full-length mRNA and CDS of the tomato MYB homolog (Solyc04g008870.2.1) were retrieved from the Sol Genomics Network (https://solgenomics.net/ (accessed on 5 March 2020)) database. To identify the *SlMYBS2* gene, the full-length CDS (808 bp) of *SlMYBS2* was cloned via PCR (30 cycles of 98 °C for 10 s, 60 °C for 5 s, and 68 °C for 5 s kb^−1^) using specific primers designed using Primer 6.0 software. The PCR product was inserted into pCaMV (under the control of the 35S CaMV promoter). Sequencing of positive clones was performed by inserting PCR products into pCaMV. All the primers used in the study are shown in Appendix A.

The *SlMYBS2* sequence was examined by checking the NCBI Conserved Domain Database (CDD) (https://www.ncbi.nlm.nih.gov/Structure/cdd/wrpsb.cgi (accessed on 8 January 2020)). A phylogenetic tree was constructed by the neighbor-joining (NJ) method in MEGA X.

### 4.3. Subcellular Localization Vector Construction

The subcellular localization pYBA1132 with a GFP tag at the C-terminus under the control of the 35S CaMV promoter plasmid was digested with *Eco*RI and *Sal*I, and the recovered product was ligated with the recovered product of the *SlMYBS2* target fragment using DNA ligase (constitutive plasmid pYBA1132-SlMYBS2-GFP). The ligation product was transformed into *Escherichia coli*, and a resistant plate was used. The primers SlMYBS2-EcoRI and SlMYBS2-SalI were used to screen for positive clones by PCR amplification. One positive clone was selected for sequencing. The pYBA1132-SlMYBS2-GFP plasmid was transformed into *Agrobacterium* GV3101 via electroporation. A single colony was selected and shaken in solution overnight at 28 °C. The bacterial solution was centrifuged at 4000× *g* rpm for 5 min and suspended in infection solution (containing 10 mM MgCl_2_, 50 mM MES (pH 5.6), and 100 µM acetyleugenone), after which the OD value was adjusted to 1–1.5. One-month-old *A. thaliana* and *Nicotiana benthamiana* leaves were injected and cultured in the dark for 48 h at 25 °C. Fluorescence signals were observed by confocal microscopy (Leica, Wetzlar, Germany).

### 4.4. Analysis of Transcriptional Activation in Yeast Cells

The target fragment and pGBKT7 vector were digested with EcoRI and BamHI and then ligated and transformed into *Escherichia coli*. The Y2Hgold yeast strain was transformed after correct digestion. The yeast strain transformed into pGBKT7 was used as the control. The transformed yeast solution was diluted by 1×, 10×, and 100×. Then, 10 µL of the solution was dropped onto tryptophan-deficient medium (SD/−Trp); tryptophan, adenine, and histidine-deficient medium (SD/−Trp− Ade− His); and the same medium supplemented with 5-bromo-4-chloro-3-indole-ad-galactoside (SD/−Trp−Ade−His + X-a-gal) plates. The samples were cultured at 28 °C for 48–96 h for observation and imaging.

### 4.5. Vector Construction and Plant Transformation

Vector construction was performed as previously described [67]. CRISPR-P (http://cbi.hzau.edu.cn/crispr/ (accessed on 12 March 2020)) was used to select specific single guide RNAs (sgRNAs) that targeted *SlMYBS2*. The targets cloned into the pYLCRISPR/Cas9 vector were named pYLCRISPR/Cas9-SlMYBS2. Using the *Agrobacterium*-mediated transformation method, pYLCRISPR/Cas9-SlMYBS2 plasmids were transformed into “Alisa Craig” according to previously described methods [68]. The transgenic tomato lines were selected based on kanamycin resistance. The primers used for vector construction are listed in Appendix A.

### 4.6. Total RNA Extraction and cDNA Synthesis

Total RNA was extracted from 4-week-old tomato leaves using a previously described TRIzol method. A TransScript II One-Step gDNA Removal and cDNA Synthesis kit (TransGen Biotech, Beijing, China) was used to synthesize cDNA. The cDNA and RNA samples obtained were stored at −80 °C until use.

### 4.7. qRT-PCR Analysis

The target sequences of the genes were amplified using specific PCR primers designed using NCBI (https://www.ncbi.nlm.nih.gov/tools/primer-blast/ (accessed on 12 February 2021)) and synthesized by the TSINGKE Institute. The qRT-PCR system consisted of 10 μL of SYBR^®^ Green Real-time PCR Master Mix, 0.8 μL of forward/reverse primers, 2 μL of cDNA template, and ddH_2_O to a total volume of 20 μL. The qRT-PCR program was as follows: 40 cycles of 95 °C for 15 s, 60 °C for 15 s, and 72 °C for 45 s. For gene expression analyses, qRT-PCR was performed with 3 independent biological replicates. *EF1α* served as a reference gene [69]. The relative expression levels were calculated using the 2^−^^ΔΔ^^CT^ method [70].

### 4.8. Observation of Stained Tissue

At 3 and 5 days post inoculation (dpi), the leaves were stained with a 0.1% trypan blue solution [71] to observe the hypersensitive response (HR). The accumulation of H_2_O_2_ and O^2^^−^ in *SlMYBS2* knockout and control plant leaves was detected by 3,3′-diaminobenzidine (DAB) and nitrotetrazolium blue chloride (NBT) staining [72] at 0–4 dpi, respectively. Ten leaves were picked for each experiment.

Cell death was observed by TB staining, with destaining in Farmer’s solution (95% ethanol/chloroform/acetic acid at a volumetric ratio of 6:3:1) for 3 h and boiling in 0.1% trypan blue solution at 65 °C for 2 h, followed by transfer to a saturated chloral hydrate solution for 4 h. The leaves were ultimately observed under a light microscope.

The production of H_2_O_2_ and O^2^^−^ was detected via DAB and NBT staining. Infected tomato leaves were incubated in 0.1% DAB and NBT solution at room temperature in the dark for 12 h and then boiled in a 96% ethanol solution for 10 min. The leaves were ultimately observed under a light microscope.

### 4.9. Statistical Analyses

All experiments were repeated independently three times. Data obtained from three independent experiments were subjected to statistical analysis according to Student’s *t*-test, and probability values of *p* ≤ 0.05 were considered significant.

## 5. Conclusions

*SlMYBS2* is a member of the MYB family and is induced by exogenous hormones (SA, JA). Compared with the wild-type plants, the mutant plants exhibited increased ROS contents, decreased expression levels of resistance genes, and more severe disease. Thus, *SlMYBS2* positively regulates resistance to *P. infestans*.

## Figures and Tables

**Figure 1 ijms-22-11423-f001:**
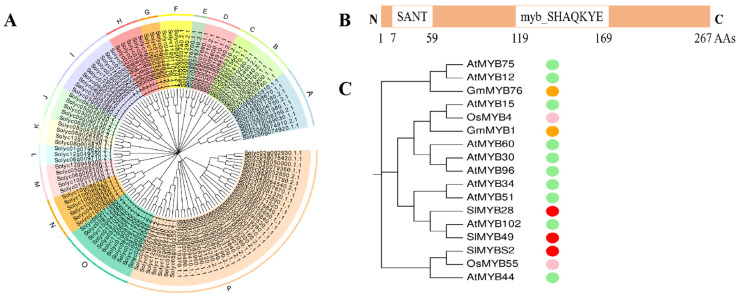
(**A**) Neighbor-joining (NJ) phylogenetic tree of SlR2R3MYB family genes. The unrooted phylogenetic tree from the complete protein sequence was depicted by the MEGA X program with the NJ method. The tree shows the 16 phylogenetic subgroups (A–P) with high bootstrap values. (**B**) Protein domain structure of *SlMYBS2*. (**C**) Phylogenetic analysis of MYB homologs in different species. Red circle, tomato; green circle, Arabidopsis; pink circle, rice; orange circle, soybean.

**Figure 2 ijms-22-11423-f002:**
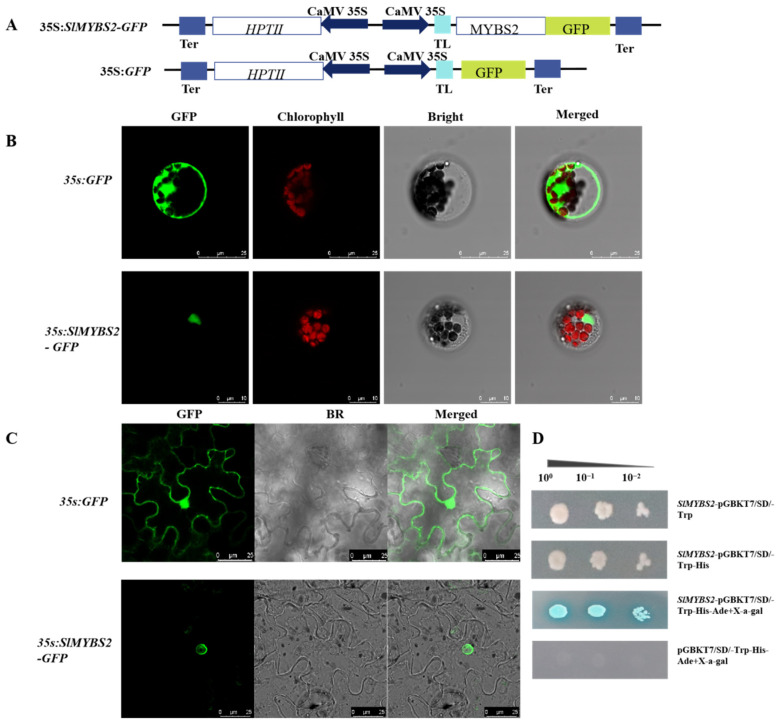
Subcellular localization and transcriptional activity analyses of *SlMYBS2*. (**A**) Schematic diagrams of the T-DNA regions of constructs used in the subcellular localization assay. *HPTII*, hygromycin-resistant gene. (**B**,**C**) Localization of *SlMYBS2* in *A. thaliana* and *Nicotiana benthamiana* cells (35S:GFP as the control group, 35S:*SlMYBS2*-GFP as the experimental group). In each panel, the extreme left box shows GFP fluorescence, the second box shows chlorophyll fluorescence, and the third bright-field box and the right box show an overlay of the two images, as indicated at the top of the image. (**D**) Analysis of the transcriptional activity of *SlMYBS2* using a yeast system.

**Figure 3 ijms-22-11423-f003:**
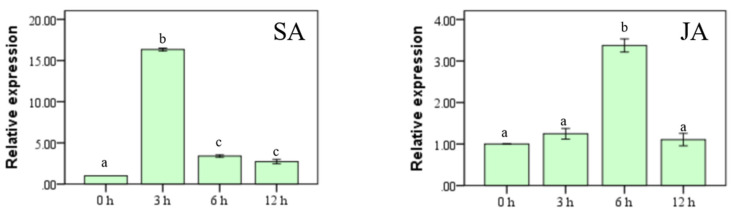
Expression patterns of *SlMYBS2* in response to defense signaling hormones. Tomato plants were treated by foliar spraying of 400 μM SA and 200 μM JA. Gene expression was analyzed by qRT-PCR, and the relative expression levels were calculated by comparison with the corresponding values at 0 h (as a control) after treatment. Relative expression is shown as the fold change in the actin transcript values. The data are presented as the means ± SDs from three independent experiments, and the different lowercase letters indicating significant differences at the 0.05 level.

**Figure 4 ijms-22-11423-f004:**
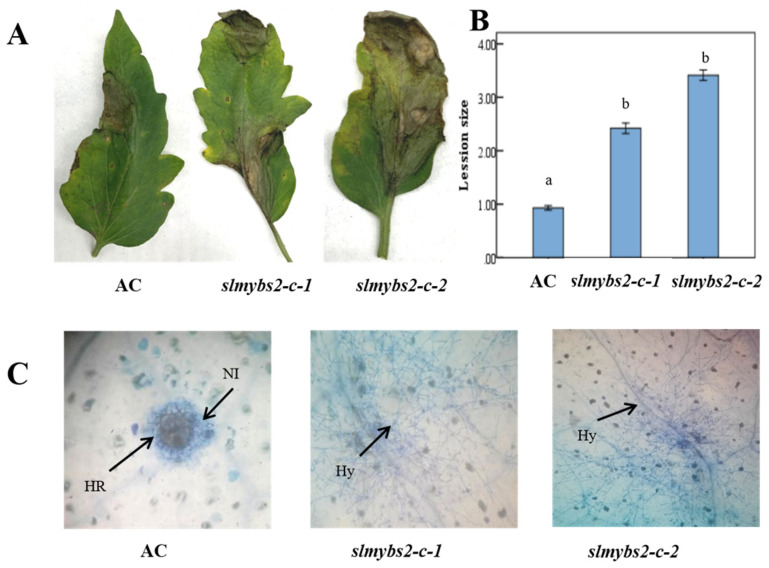
Knocking out *SlMYBS2* reduced the resistance to *P.*
*infestans*. Infiltration of *slmybs2-c-* and AC plants with *P. infestans.* (**A**–**C**) Parameters showing the extent of the disease five days after infection with *P. infestans*. (**A**) Disease symptoms; (**B**) lesion size (cm); (**C**) number of necrotic cells (scale bars: 100 μm). Hy, hyphae; Nl, necrotic lesions; HR, hypersensitive response. Different lowercase letters indicating significant differences at the 0.05 level.

**Figure 5 ijms-22-11423-f005:**
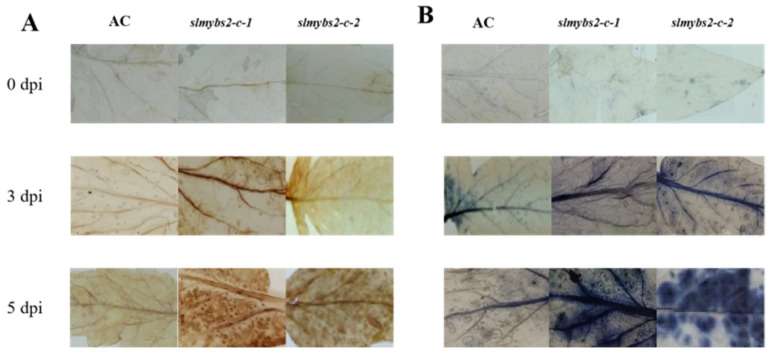
Accumulation of H_2_O_2_ (**A**) and O^2−^ (**B**) in AC and *slmybs2-c*-plants after inoculation with *P. infestans.* (**A**) Accumulation of H_2_O_2_. Tomato leaves were stained with DAB at 0, 3, and 5 dpi. (**B**) Accumulation of O^2−^ in tomato leaves stained with NBT at 0, 3, and 5 dpi.

**Figure 6 ijms-22-11423-f006:**
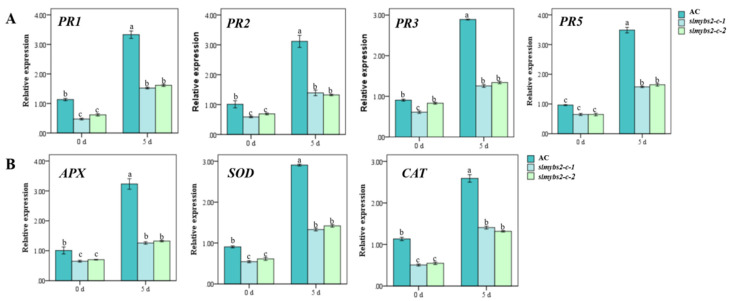
Knocking out *SlMYBS2* affected the expression of ROS generation- and scavenging-related and defense-mediated genes after infection with *P. infestans*. (**A**) Expression of defense-related genes(*PR1, PR2, PR3, PR5*). (**B**) Expression of ROS generation- and scavenging-related genes (*APX, SOD, CAT*). Relative expression levels are shown as fold changes in actin transcript values. The data are presented as the means ± SDs from three independent experiments, and the different lowercase letters indicating significant differences at the 0.05 level.

**Figure 7 ijms-22-11423-f007:**
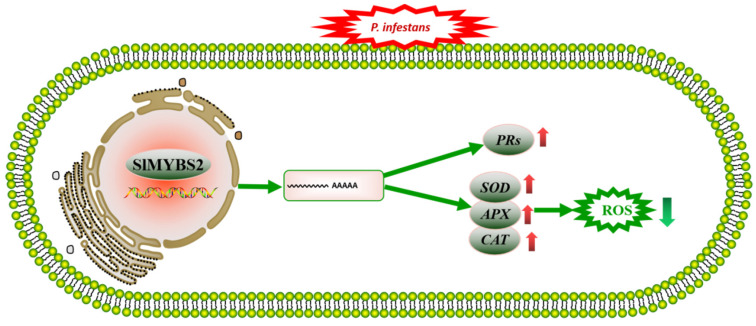
A proposed model for the putative role of *SlMYBS2* in the regulation of pathogen defense responses. The green arrows pointing upward represent upregulated genes or functions, while the red arrows pointing downward represent downregulated genes or functions. The green solid lines represent our results.

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
