# Peer review of "CRISPR/Cas9-Mediated SlMYBS2 Mutagenesis Reduces Tomato Resistance to Phytophthora infestans"

_ijms, 2021, doi:10.3390/ijms222111423_

Round 1
Reviewer 1 Report
Liu et al. show in this study that a tomato transcription factor, SlMYBS2 is induced by defense-related hormones (SA, JA) and is necessary for resistance to Phytophthora infestans. Compared to wild-type plants, CRISPR/Cas9-generated mutant (slmybs2-c) plants exhibit an impaired resistance (i.e. increased susceptibility) to P. infestans associated with increased ROS contents, and down-regulation of PR and antioxidant gene expression. These results are relevant, especially from the perspective of pest management, since Phytophthora diseases and the damage they cause are very difficult to control, therefore every piece of information on plant defense mechanisms may help the process of resistance breeding.
However, some problems in the MS worth mentioning and should be resolved.
- In the second paragraph of Introduction you write that „MYB30 and MYB96 respond to pathogen stress by regulating the expression of corresponding genes in Arabidopsis” - What are these corresponding genes? From the papers cited here (Li et al., 2009 and Seo et al., 2010) these are likely genes associated with hormone-mediated signaling and hormone biosynthesis. Please specify.
- A few lines below in the Intro you write that in rice OsMYB55 improved tolerance to high temparatures and "OsMYB55 plays a certain role under low, rather than high, temperature stress ". Does this mean that OsMYB55 also has a role in tolerance to low temperatures? Please specify.
- In the final paragraph of Introduction You write here that "we previously showed that SlMYBS2 was induced by a late blight pathogen"- however, the relevant citation is missing.
- In Materials and methods, Lx (Lux) is probably not the best unit to describe light intensity in a plant growth chamber - it is more accepted to use micromoles per square m per sec.
- Still in Methods, you write that „The tomato plants were sprayed with the same amount of 400 μM salicylic acid (SA) and 200 μM methyljasmonate (MeJA)” – exactly what amounts of 400 uM SA and 200 uM MeJA were used for spraying tomato plants?
- In the method section describing gene cloning, I suppose that "reaction protocol" for SlMYBS2 cloning was a PCR protocol - please clarify.
-In methods, section 2.3: Restriction enzymes are spelled with the first part (referring to the organism of origin, e.g. Eco or Sal) in italics.
- There is a serious error that shows up several times in Results: Change "knocking down" to "knocking out" - you used CRISPR mutagenesis to permanently change the sequence of SlMYBS2...(see e.g. the Abstract text, where this is correctly stated).
- In Results, the title of section 3.6: „ROS accumulation was decreased in slmybs2 mutant plants…”. This MUST be changed: ROS accumulation increased in the slmybs2 mutants upon infection (indicating their susceptibility to P. infestans). In fact, this is correctly stated elsewhere (in the Abstract and the bottom of this paragraph).
- Legend of Fig. 5: For superoxide detection, leaves were stained with NBT, not DAB...! (see in Methods)
- In Discussion section 4.2 you write that „NPR1 is …located upstream of SA biosynthesis”. However, NPR1 is located downstream of SA biosynthesis - it has been shown that NPR1 is an SA-responsive transcriptional coactivator, see e.g. Mou et al. 2003; Cell 113:935–944 Dong 2004; Curr Opin Plant Biol 7:547–552 Tada et al.2008 Science 321:952–956.
- In Discussion section 4.4, instead "semi-living trophic" the correct term is "hemibiotrophic" and instead "dead-nutrient" the correct term here is "necrotrophic".
- Fig. 7 is a model „presenting the putative role of SlMYBS2 in the regulation of pathogen defense responses”, based on results of this study. However, the Fig. 7 graph depicts effects of the mutant (slmybs2-c), therefore, it should be corrected, since you showed that SlMYBS2 has a role in downregulating ROS and upregulating antioxidants (SOD, APX and CAT) and PR genes. Please correct Fig. 7 accordingly.
Author Response
Liu et al. show in this study that a tomato transcription factor, SlMYBS2 is induced by defense-related hormones (SA, JA) and is necessary for resistance to Phytophthora infestans. Compared to wild-type plants, CRISPR/Cas9-generated mutant (slmybs2-c) plants exhibit an impaired resistance (i.e. increased susceptibility) to P. infestans associated with increased ROS contents, and down-regulation of PR and antioxidant gene expression. These results are relevant, especially from the perspective of pest management, since Phytophthora diseases and the damage they cause are very difficult to control, therefore every piece of information on plant defense mechanisms may help the process of resistance breeding.
However, some problems in the MS worth mentioning and should be resolved.
- In the second paragraph of Introduction you write that „MYB30 and MYB96 respond to pathogen stress by regulating the expression of corresponding genes in Arabidopsis” - What are these corresponding genes? From the papers cited here (Li et al., 2009 and Seo et al., 2010) these are likely genes associated with hormone-mediated signaling and hormone biosynthesis. Please specify.
Thank you very much for your responsible comments.
These corresponding genes are PR genes. And these are likely genes associated with hormone-mediated signaling and hormone biosynthesis.
I have changed the sentence in the article to “MYB30 and MYB96 respond to pathogen stress by regulating the expression of corresponding PR genes in Arabidopsi. ”
- A few lines below in the Intro you write that in rice OsMYB55 improved tolerance to high temparatures and "OsMYB55 plays a certain role under low, rather than high, temperature stress ". Does this mean that OsMYB55 also has a role in tolerance to low temperatures? Please specify.
Thank you very much for your kindly advice.
I’m very sorry, but we made a mistake in this sentence. We have corrected the wrong sentence. “The overexpression of OsMYB55 in rice increased the total amino acid content in the plant, which improved plant tolerance to high temperature. OsMYB55 plays a certain role under high temperature stress”
- In the final paragraph of Introduction You write here that "we previously showed that SlMYBS2 was induced by a late blight pathogen"- however, the relevant citation is missing.
Thank you very much for your kind advice. Since the reference transcriptome article has not yet been published, it is not convenient to insert references.
- In Materials and methods, Lx (Lux) is probably not the best unit to describe light intensity in a plant growth chamber - it is more accepted to use micromoles per square m per sec.
Thank you very much for your suggestions.
We have changed the unit to μmol m-2 S-1 according to your valuable suggestion.
- Still in Methods, you write that „The tomato plants were sprayed with the same amount of 400 μM salicylic acid (SA) and 200 μM methyljasmonate (MeJA)” - exactly what amounts of 400 uM SA and 200 uM MeJA were used for spraying tomato plants?
Thank you very much for your kind advice.
Spray tomato plants with 5ml of 400uMSA and 200uMMeJA respectively.
Modified in the article according to your suggestion
- In the method section describing gene cloning, I suppose that "reaction protocol" for SlMYBS2 cloning was a PCR protocol - please clarify.
Thank you very much for your responsible comments.
We have changed the sentence to “The PCR product is inserted into pCaMV (under the control of the 35S CaMV promoter)”
-In methods, section 2.3: Restriction enzymes are spelled with the first part (referring to the organism of origin, e.g. Eco or Sal) in italics.
Thank you very much for your responsible comments. The “EcoRI and SalI” of the article has been changed to “EcoRI and SalI”.
- There is a serious error that shows up several times in Results: Change "knocking down" to "knocking out" - you used CRISPR mutagenesis to permanently change the sequence of SlMYBS2...(see e.g. the Abstract text, where this is correctly stated).
Thank you very much for your kind comments.
According to your valuable suggestions, all the wrong writing methods in the article have been corrected.
- In Results, the title of section 3.6: „ROS accumulation was decreased in slmybs2 mutant plants…”. This MUST be changed: ROS accumulation increased in the slmybs2 mutants upon infection (indicating their susceptibility to P. infestans). In fact, this is correctly stated elsewhere (in the Abstract and the bottom of this paragraph).
I'm really sorry for making such a stupid mistake,and thank you very much for your valuable comments. It has been changed to the correct title in the article.
“ROS accumulation was increased in slmybs2 mutant plants compared to control plants after inoculation with P. infestans”
- Legend of Fig. 5: For superoxide detection, leaves were stained with NBT, not DAB...! (see in Methods)
Thank you very much for your kind comments. It has been changed to the correct sentence in the text: “Accumulation of O2- in tomato leaves stained with NBT at 0, 3, and 5 dpi.”
- In Discussion section 4.2 you write that „NPR1 is …located upstream of SA biosynthesis”. However, NPR1 is located downstream of SA biosynthesis - it has been shown that NPR1 is an SA-responsive transcriptional coactivator, see e.g. Mou et al. 2003; Cell 113:935-944 Dong 2004; Curr Opin Plant Biol 7:547-552 Tada et al.2008 Science 321:952-956.
Thank you very much for your suggestion.
It has been changed to the correct sentence based on your opinion and highlighted in red.
- In Discussion section 4.4, instead "semi-living trophic" the correct term is "hemibiotrophic" and instead "dead-nutrient" the correct term here is "necrotrophic".
Thank you very much for your kindly advice. In the article, "semi-living trophic" has been changed to "hemibiotrophic", and "dead-nutrient" has been changed to "necrotrophic".
- Fig. 7 is a model „presenting the putative role of SlMYBS2 in the regulation of pathogen defense responses”, based on results of this study. However, the Fig. 7 graph depicts effects of the mutant (slmybs2-c), therefore, it should be corrected, since you showed that SlMYBS2 has a role in downregulating ROS and upregulating antioxidants (SOD, APX and CAT) and PR genes. Please correct Fig. 7 accordingly.
Thank you very much for your suggestion. The model diagram has been modified according to your suggestions.

Reviewer 2 Report
In this study, the authors reveal the involvement of the SlMYBS2 gene in tomato resistance to Phytophthora infestans by knocking out the gene through CRISPR-Cas9 technology. The results indicate that SlMYBS2 acts as a positive regulator of tomato resistance to P. infestans infection by regulating the ROS level and the expression of pathogenesis-related genes. To my opinion, the results obtained are interesting. However, there are several major concerns that have to be addressed before the final acceptance of the manuscript.
1. Do not use abbreviation in the title. Instead of P. infestans, use Phytophthora infestans.
2. In the end of the abstract is mentioned “by alleviating cell membrane injury” but such data are not provided in the manuscript.
3. Introduction – This part would greatly benefit from the editorial service or some support from an expert or more senior colleagues that could provide help with the English language and style.
- References do not follow the journal’s instructions. They must be NUMBERED in order of appearance in the text, and the reference numbers should be placed in square brackets listed at the end of the manuscript.
- I would suggest some changes in the way of citing references over the introduction section, for instance:
- instead of ”Paz-Ares was the first to clone a MYB transcription factor in maize that regulated the synthesis of maize anthocyanins (Paz-Ares J et al., 1987), use “Paz-Ares et al. (1987) were the first to clone a MYB transcription factor that regulated the synthesis of maize anthocyanins”;
- instead of “Li used virus-induced gene silencing to silence SlMYB28 in tomato plants to thereby improve resistance to the tomato yellow leaf curl virus (TYLCV) (Li et al., 2018)”, use “To improve resistance of tomato plants to the yellow leaf curl virus (TYLCV), virus-induced gene silencing of SlMYB28 has been used (Li et al., 2018)”;
- instead of “Li (2018) used CRISPR/Cas9 technology to edit five genes that improved the habitat of tomato plants, advanced the flowering time, and in-creased the size of tomato fruit and the content of vitamin C in the tomatoes (Li TD et al., 2018)”, rephrase as “Li et al. (2018) used CRISPR/Cas9 technology to edit five genes that improved the habitat of tomato plants, advanced the flowering time, and increased the size of tomato fruit and the content of vitamin C in the tomatoes.”;
- instead of “Soky knocked out the flowering suppressor genes SELF-PRUNING 5G and SP5G in tomato using CRISPR/Cas9, which induced an earlier flowering period for the edited seedlings and improved the yield to a certain extent (Soyk S et al., 2016).”, use “The flowering suppressor gene SELF-PRUNING 5G (SP5G) in tomato have been knocked out by CRISPR/Cas9, which induced earlier flowering and yield of the edited seedlings (Soyk et al., 2016).” This publication deals with SELF-PRUNING 5G, abbreviated as SP5G – this is a single gene, not two genes, as it was wrongly stated in the text.
- “Good resistance genes” (end of page 2) – What does this mean
4. The Materials and Methods (M&M) section requires substantial improvement. The authors must provide more clear explanation of their experimental procedures and be more specific.
- “Seedlings of tomato (Alisa Craig and the mutant plants)” – please justify what is Alisa Craig (a variety, cultivar or..), and which mutant plants have been used (not only “mutant plants”).
- All plant species are grown at 24°C (first paragraph of M&M), then the infected plants are growth at a temperature of 28°C. This is a quite large temperature difference that could affect plant growth and development. Why the growth temperature was changed?
- Not very clear experimental setup – “detached tomato leaves were inoculated”, then the authors describe 15 control and mutant plants.
- “leaves were harvested at specific days post inoculation” - which days?
- 2.2. Gene cloning and bioinformatic analysis – What is the size of the cloned cDNA fragment?
- “A partCAM-SlMYBS2 vector was constructed for the identification of positive clones” – Please specify.
- 2.3. Subcellular localization vector construction. “One-month-old and four-week-old tobacco leaves...” – What is the difference?
- “Fluorescence signals were observed by confocal microscopy.” – Please provide information about the fluorescence tag?
- 2.4. Analysis of transcriptional activation in yeast cells - “Then, 10 μL of the solution was dropped onto SD/−Trp, SD/−Trp− Ade− His and SD/−Trp−Ade−His+X-a-gal plates”. Introduce abbreviations that are currently provided in page 6, e.g. tryptophan-deficient medium (SD/−Trp), tryptophan, adenine and histidine (SD/−Trp−Ade−His), etc.
-2.7. qRT-PCR analysis - Were the data normalised with a single gene or EFα1 was selected as a representative gene? This needs to be specified in methods as well as in figure legends.
-Correct spelling of superoxide anion is not O2- but O2−
5. Results
-Second paragraph “Based on our previous pathogen-induced transcriptome analysis..” – provide reference
-Mutants and gene symbols must be italicized (page 7)
3.6. ROS accumulation… - Move the first sentence to discussion.
-In the legends of Figure 3 and Figure 6, introduce abbreviations that are used in the graphs, e.g. in Fig. 3: SA and JA; in Fig.6: APX, SOD, CAT, and all PR genes.
6. Discussion - This section needs substantial revision with more critical discussion of the results obtained.
- The first two headings in the discussion must be more precisely defined. The first one is entitled “CRISPR/Cas9 technology was used to edit out the SlMYBS2 gene in tomato”. However, most of the comments overlap with introduction. Only the last two sentences mention CRISPR/Cas9 technology as a tool used in this study, but there is no a real discussion. Maybe this part should rather focusses on the application potentials of CRISPR/Cas9 genome editing for studying biotic stress in tomato. The second paragraph also does not properly discuss the regulation of SlMYBS2 by phytohormones. A very brief discussion of the induction of MYB transcription factors is only provided in the paragraph end.
-Some of the contents in next paragraphs mostly repeat the results obtained. Can the authors try to explore the results in more depth and go into greater detail about their meaning and implication.
-Rephrase the very last sentence – “weakened scavenging ability of ROS” is not the right way to speak about ROS scavenging ability of antioxidant machinery.
Author Response
Dear Editor,
Here is our revised manuscript ijms-1412853 for “International Journal of Molecular Sciences”. Response to the comments:
Reviewer #2:
Comments and Suggestions for Authors
In this study, the authors reveal the involvement of the SlMYBS2 gene in tomato resistance to Phytophthora infestans by knocking out the gene through CRISPR-Cas9 technology. The results indicate that SlMYBS2 acts as a positive regulator of tomato resistance to P. infestans infection by regulating the ROS level and the expression of pathogenesis-related genes. To my opinion, the results obtained are interesting. However, there are several major concerns that have to be addressed before the final acceptance of the manuscript.
- Do not use abbreviation in the title. Instead of P. infestans, use Phytophthora infestans.
Thank you very much for your responsible comments. In the title, “P. infestans” has been changed to “Phytophthora infestans.”
- In the end of the abstract is mentioned “by alleviating cell membrane injury” but such data are not provided in the manuscript.
Thank you very much for your kindly suggestion.
Due to the accumulation of reactive oxygen species, it will cause toxic effects on plant cells, leading to membrane lipid peroxidation, cell membrane damage, and bacteria susceptible to cell death. (Sathiyaraj G., Lee O. R., Parvin S., Khorolragchaa A., Kim Y. J. and Yang D. C.. Transcript profiling of antioxidant genes during biotic and abiotic stresses in Panax ginseng CA Meyer. Mol Biol Rep., 2011, 38:2761-2769 ). However, referring to the reviewer’s suggestions, there is really no relevant data to support this article. It is inaccurate to put this sentence in the abstract. It is my mistake, so this sentence has been deleted from the abstract
- Introduction – This part would greatly benefit from the editorial service or some support from an expert or more senior colleagues that could provide help with the English language and style.
- References do not follow the journal’s instructions. They must be NUMBERED in order of appearance in the text, and the reference numbers should be placed in square brackets listed at the end of the manuscript.
The format of the references has been modified according to the requirements of the journal
- I would suggest some changes in the way of citing references over the introduction section, for instance:
Thank you very much for your suggestion.
This section was modified according to your kind suggestion.
“Good resistance genes” (end of page 2) – What does this mean
Thank you very much for your comments. What I want to express is a gene with higher resistance to P. infestans. This sentence has been revised in the article and marked in red
“the identification of higher resistance genes to late blight is urgent.”
- The Materials and Methods (M&M) section requires substantial improvement. The authors must provide more clear explanation of their experimental procedures and be more specific.
Thank you very much for your responsible comments.
- “Seedlings of tomato (Alisa Craig and the mutant plants)” – please justify what is Alisa Craig (a variety, cultivar or..), and which mutant plants have been used (not only “mutant plants”).
Thank you very much for your kind comments. Alisa Craig is a wild type tomato variety. This part has been supplemented with more specific plant materials based on your suggestions.
“Tomato (Solanum lycopersicum cv. Ailsa Craig) including wild-type and two mutant-type (slmybs2-c-1, slmybs2-c-2)”
- All plant species are grown at 24°C (first paragraph of M&M), then the infected plants are growth at a temperature of 28°C. This is a quite large temperature difference that could affect plant growth and development. Why the growth temperature was changed?
Thank you very much for your meaningful questions. We made this part wrong.
In fact, the plants after inoculation are grown at 21°C. Compared with 24℃, there is still a certain temperature difference. However, since the optimum temperature for late blight to infect plants is 21°C, the inoculated plants are cultured at 21°C. And the suitable growth temperature for tomatoes is 15-30℃
- Not very clear experimental setup – “detached tomato leaves were inoculated”, then the authors describe 15 control and mutant plants.
Thank you very much for your kindly suggestion.
According to your valuable suggestions, this part has been written more specifically, and it is marked in red in the text
“A spore suspension of P. infestans (1×106 spores/mL of water) was used to inoculate detached tomato leaves(Six leaves of control plants and mutant plants) and tomato plants(Fifteen control and mutant plants)were infected by spraying. ”
- “leaves were harvested at specific days post inoculation” - which days?
Thank you very much for your sincere suggestions.
“ leaves were harvested at 0-5 days post inoculation (dpi) for further analysis”
This sentence has also been revised in the article.
- 2.2. Gene cloning and bioinformatic analysis – What is the size of the cloned cDNA fragment?
Thank you very much for your kind advice.
The size of the cioned cDNA fragment is 808bp.
- “A partCAM-SlMYBS2 vector was constructed for the identification of positive clones” – Please specify
Thank you very much for your responsible comments. According to your suggestion, the test method of this part has been revised.
“In order to identify the SlMYBS2 gene, the full-length CDS of SlMYBS2 was cloned via PCR(30 cycles of 98°C for 10 s, 60°C for 5 s, and 68°C for 5 s kb−1) using specific primers designed via Primer 6.0 software. The PCR product is inserted into pCaMV (under the control of the 35S CaMV promoter) .Sequencing of positive clones obtained by inserting PCR products into pCaMV. ”
- 2.3. Subcellular localization vector construction. “One-month-old and four-week-old tobacco leaves...” – What is the difference?
Thank you very much for your suggestion. Thank you very much for your question, this part is our mistake, and I am very sorry for that. The correct writing is now shown in the article.
“ One-month-old A. thaliana and Nicotiana benthamiana leaves were injected ...”
- “Fluorescence signals were observed by confocal microscopy.” – Please provide information about the fluorescence tag?
This part has been modified based on your valuable comments and marked in red
“The subcellular localization pYBA1132 with a GFP tag at the C-terminus under the control of the 35S CaMV promoter”
- 2.4. Analysis of transcriptional activation in yeast cells - “Then, 10 μL of the solution was dropped onto SD/−Trp, SD/−Trp− Ade− His and SD/−Trp−Ade−His+X-a-gal plates”. Introduce abbreviations that are currently provided in page 6, e.g. tryptophan-deficient medium (SD/−Trp), tryptophan, adenine and histidine (SD/−Trp−Ade−His), etc.
Thank you very much for your valuable comments. The text has been modified according to your suggestion.
“Then, 10 µL of the solution was dropped onto tryptophan-deficient medium (SD/−Trp), tryptophan, adenine and histidine-deficient medium (SD/−Trp− Ade− His) and the same medium supplemented with 5-bromo-4-chloro-3-indole-ad-galactoside(SD/−Trp−Ade−His+X-a-gal)plates”
-2.7. qRT-PCR analysis - Were the data normalised with a single gene or EFα1 was selected as a representative gene? This needs to be specified in methods as well as in figure legends.
Dear reviewer, I'm sorry I'm not sure about the expression.
EF1α serving as a reference gene. We have revised this sentence in the manuscript
Rotenberg D, Thompson TS, German TL, et al. Methods for effective realtime RT-PCR analysis of virus-induced gene silencing. J Virol Methods. 2006;
138(1–2):49–59.
Livak KJ, Schmittgen TD. Analysis of relative gene expression data using realtime quantitative PCR and the 2-△△CT method. Methods. 2001;25:402–8.
-Correct spelling of superoxide anion is not O2- but O2−
Thank you very much for your question. We have corrected the wrong writing in the manuscript.
- Results
-Second paragraph “Based on our previous pathogen-induced transcriptome analysis..” – provide reference
Thank you very much for your kind advice. Since the reference transcriptome article has not yet been published, it is not convenient to insert references.
-Mutants and gene symbols must be italicized (page 7)
According to your valuable comments, the wrong writing in the article has been corrected
3.6. ROS accumulation… - Move the first sentence to discussion.
Thank you very much for your sincere suggestions, I have put this paragraph in the discussion section in the article
-In the legends of Figure 3 and Figure 6, introduce abbreviations that are used in the graphs, e.g. in Fig. 3: SA and JA; in Fig.6: APX, SOD, CAT, and all PR genes.
Thank you very much for your sincere suggestions. This part has been modified based on your valuable comments and marked in red.
- Discussion - This section needs substantial revision with more critical discussion of the results obtained.
- The first two headings in the discussion must be more precisely defined. The first one is entitled “CRISPR/Cas9 technology was used to edit out the SlMYBS2 gene in tomato”. However, most of the comments overlap with introduction. Only the last two sentences mention CRISPR/Cas9 technology as a tool used in this study, but there is no a real discussion. Maybe this part should rather focusses on the application potentials of CRISPR/Cas9 genome editing for studying biotic stress in tomato. The second paragraph also does not properly discuss the regulation of SlMYBS2 by phytohormones. A very brief discussion of the induction of MYB transcription factors is only provided in the paragraph end.
-Some of the contents in next paragraphs mostly repeat the results obtained. Can the authors try to explore the results in more depth and go into greater detail about their meaning and implication.
-Rephrase the very last sentence – “weakened scavenging ability of ROS” is not the right way to speak about ROS scavenging ability of antioxidant machinery
Thank you very much for your valuable suggestions. I have worked hard to make a lot of changes in the discussion part, and the changes are marked in red. If you still have any questions, please feel free to let me know, and I will work harder to modify it and make it better.
Finally, the manuscript has been revised according to comments of reviewers. All changes in the manuscript were marked by red color. In addition, the American Journal Experts was used for English modification. If there is any shortage, please inform us! We will try our best to revise this manuscript.
Thank you very much for your attention and kind advice.
